# Pilot Study on Risk Perception in Practices with Medical Cyclotrons in Radiopharmaceutical Centers in Latin American Countries: Diagnosis and Corrective Measures

**DOI:** 10.3390/ijerph22121885

**Published:** 2025-12-18

**Authors:** Frank Montero-Díaz, Antonio Torres-Valle, Ulises Jauregui-Haza

**Affiliations:** 1Area of Basic and Environmental Science, Instituto Tecnológico de Santo Domingo (INTEC), Santo Domingo 10602, Dominican Republic; ulises.jauregui@intec.edu.do; 2Instituto Nacional del Cáncer Rosa Emilia Sánchez Pérez de Tavares (INCART), Santo Domingo 10602, Dominican Republic; 3Higher Institute of Technologies and Applied Sciences (INSTEC), Universidad de la Habana, La Habana 10100, Cuba; atorres@instec.cu; 4Cátedra UNESCO de Cambio Climático, Resiliencia y Sistemas Complejos, Instituto Tecnológico de Santo Domingo (INTEC), Santo Domingo 10602, Dominican Republic

**Keywords:** cyclotron, sustainable development, risk perception, variables, surveys

## Abstract

Practices with medical cyclotrons to produce PET radiopharmaceuticals in Latin America represent a technological advance for the diagnosis and treatment of diseases such as cancer, but they involve occupational risks due to exposure to ionizing radiation. This study evaluates the perception of risk in 46 radiopharmacy service workers in 13 countries in the region (Argentina, Bolivia, Brazil, Chile, Colombia, Costa Rica, Cuba, Ecuador, Mexico, Peru, Portugal, Dominican Republic and Venezuela), analyzing differences by gender and age. The questionnaire, validated by reliability analysis (Cronbach’s coefficient α > 0.7), was statistically analyzed with means, standard deviations (SD) and standard errors (SE), 95% confidence intervals (Student’s t-distribution), and coefficients of variation (CV) to assess the dispersion of each variable. The results reveal general underestimation in dimensions such as reversibility of consequences (SD = 0.7142, SE = 0.1053) and familiarity (SD = 0.8410, SE = 0.124), promoting complacency, while immediacy of consequences shows overestimation (SD = 0.9760, SE = 0.1439), amplifying anxiety. By gender, women tend to overestimate (e.g., immediacy = 2.5) and men underestimate (e.g., confidence = 1.78); by age, young people (26–45 years old) overestimate more than older people (≥46 years old). These deviations, with high QoL indicating heterogeneity, suggest interventions such as continuous training, real-time monitoring, and communication campaigns to balance perception. Practical recommendations include job rotations to reduce underestimation due to familiarity and simulations to mitigate emotional overestimation, which are aligned with IAEA regulations (GSR Part 3, SSG-46) to promote a sustainable safety culture.

## 1. Introduction

Medical practices with ionizing radiation are an expression of technological progress, and, therefore, of the economic development of the world; however, such development is in conflict with environmental sustainability, given the possible impact of ionizing radiation on the environment [1]. Due to their effect on human health in the face of diseases with specialized treatment, these medical practices solve an issue of high social impact, which nevertheless conflicts with economic development because, due to their cost, these technologies are limited to countries or groups of people at a high economic level.

Occupational exposure to ionizing radiation can occur in various industries, medical institutions, teaching and research centers, and nuclear fuel cycle facilities. For the safe and regulated use of radiation, radioactive materials, and nuclear energy, appropriate levels of radiation protection for workers need to be established [2].

The production of positron emission tomography (PET) radiopharmaceuticals using cyclotrons represents a cornerstone of nuclear medicine, as it allows non-invasive imaging for the diagnosis and treatment monitoring of diseases such as cancer and neurological disorders [3,4]. Cyclotrons accelerate charged particles to produce short-lived radioisotopes, such as fluorine-18 and carbon-11, which are incorporated into radiopharmaceuticals such as [^18^F] FDG for clinical use. This process requires strict safety protocols due to the ionizing radiation involved, balancing technological advancement with occupational and environmental risks [5]. In the Latin American and Caribbean region, a total of 67 cyclotron facilities dedicated to PET radioisotope production have been identified, of which 54 are currently operational [6], highlighting the significant expansion of this technology despite ongoing challenges in radiation protection and regulatory harmonization. Although national regulatory frameworks vary, the International Atomic Energy Agency (IAEA) standards—particularly GSR Part 3 [7] and SSG-46 [8]—promote harmonization of good manufacturing practices (GMP) and radiation safety in radiopharmaceutical production across member states, providing a common technical and safety baseline for the region.

The high specificity of PET imaging improves patient outcomes, but the complexity of cyclotron operations requires robust risk management to mitigate potential human error, which is a leading cause of radiological incidents. Recent advances emphasize optimizing yields and production safety through automated synthesis modules and regulatory compliance, ensuring sustainable practices in medical isotope production [9].

According to Slovic [10], risk perception, understood as the subjective assessment of the probability and severity of consequences associated with a hazard, is crucial for workers exposed to ionizing radiation, as it affects their safety behaviors and psychological well-being.

Research on risk perception began in the 1960s. Starr [11] found that the acceptability of risk depends on the subjective scale of individuals. Slovic [10] used a psychological model to assess dimensions of risk perception. This perception reflects an understanding of objective risks, influenced by individual experience, intuitive judgment, and subjective sensation.

Risk perception involves individual assessment of situations, including describing them, estimating the ability to control risk, and determining the probability of occurrence. Based on this definition, risk perception is studied as an individual cognitive mechanism, independent of the social system [12]. However, theories at the individual level do not adequately explain how risk perception varies between or within communities. Risk perception influences safety behaviors, including safety compliance (core activities to maintain safety in the workplace) and safety engagement (behaviors that help develop a supportive safety environment) [13].

The perception of risk as a social and cultural construct goes beyond the individual level, reflecting values, history, ideology, and other factors within the social system [14]. Key factors affecting risk perception include individual differences, expectation levels, the influence of information, risk characteristics, voluntariness, and education. We can define risk perception as follows: people must recognize and feel risk in order to take protective measures, which are influenced by various social factors [15].

Risk perception in nuclear medicine is determined by cultural, emotional, and informational factors. Effective communication and education are critical to addressing misconceptions and improving patient cooperation and acceptance of procedures [16]. Similarly, risk perception can be part of the comprehensive risk analysis in the production of PET radiopharmaceuticals [17].

According to Perko [18], the greater the employee’s professional experience, both in terms of volunteerism and knowledge, the greater the perception of radiological risks. Likewise, the feeling of protection against the risks associated with nuclear installations (personal control) decreases the perception of these risks. On the other hand, greater perceived control by the authorities regarding safety in nuclear facilities (institutional control) is also associated with a lower perception of radiological risks.

On the other hand, safety culture refers to the shared values, attitudes, beliefs, and behaviors within an organization that prioritize safety over competing goals. It is the foundation of workplace behavior, ensuring that safety is constantly emphasized and integrated into daily practices. In the context of radiation safety, it encompasses scientific and societal factors, including knowledge, values, and experiences related to radiation protection, with the aim of protecting patients, workers, the public, and the environment from the harmful effects of ionizing radiation [19].

The objective of this research is to evaluate the perception of radiological risk of personnel related to medical cyclotrons in radiopharmacy services in the Latin American region and Portugal.

Individual characteristics, such as gender, age, and educational level, among others, generate significant differences in the perception of events. These attributes play a crucial role in shaping risk perception, causing the degree of risk perception to vary substantially between people. Women tend to demonstrate greater sensitivity to risk perception compared to men [20]. Older people, due to their extensive decision-making experience, tend to employ highly developed strategies to make effective decisions in real-world situations [21]. Several studies suggest a correlation between educational level and risk perception. It has been observed that an increase in radiation knowledge, derived from more advanced education, can minimize the perception of the associated risk [22].

Some studies indicate that income influences the perception of risk. For example, Zhou et al. [23] point out that higher levels of household income improve farmers’ ability to perceive risks related to soil contamination, including the perception of facts, losses, and causes.

Individuals’ perceptions of risk vary depending on their understanding of the likelihood of risk, the severity of the consequences, and the ability to control specific events. Consequently, after analyzing research from other academics, it was observed that increasing the knowledge of individuals through communication and education substantially influences their perception of danger. For example, Huang et al. [24] indicate that people who perceive themselves as more informed about the chemical industry are more likely to accept the risks associated with it. Aluko et al. [25] argued that awareness of potential occupational risks is closely linked to the formation of a positive attitude, which, in turn, informs behavior.

Recent research consistently shows that experts with higher education and professional experience have lower risk perceptions regarding nuclear waste and accidents compared to the general population. Studies show that people with greater knowledge of radiation exhibit significantly lower risk perceptions of exposure to low doses of radiation [26,27]. This pattern is evident in different expert populations, including radiological emergency medical personnel, where higher levels of knowledge were inversely correlated with risk perception [27]. The gap between experts and the public stems from differences in knowledge and understanding, with those who have little knowledge of radiation showing the highest risk perceptions [26]. Professional populations with radiation-related training demonstrate more favorable attitudes toward nuclear energy [28,29]. However, this knowledge–perception relationship can be influenced by emotional responses and underlying psychological factors [30,31]. The gap between expert and public perceptions presents ongoing challenges for risk communication in nuclear policy contexts [32,33].

Some research consistently shows that social influences significantly affect safety behaviors in high-risk occupations. Management training and commitment are crucial factors, and senior managers play an essential role in setting safety priorities and organizational culture [34,35]. However, co-workers emerge as particularly influential sources of social information and role models. Turner et al. [36] found that coworkers’ perceived support for safety was more important in keeping employees safe under high job demands, while Fugas et al. [37] demonstrated that coworkers’ descriptive safety norms significantly influenced proactive safety behaviors. McLain [38] showed that workers engage in social references when interpreting ambiguous safety hazards, looking for cues from others to guide their behavior. Although Grocutt et al. [39] found inconsistent patterns in different contexts, they confirmed that all sources of safety support (senior managers, supervisors, and co-workers) contribute to safety outcomes. Recent work emphasizes how organizational culture, group dynamics, and social norms collectively shape risk perception and compliance behaviors in hazardous work environments [40,41].

## 2. Materials and Methods

### 2.1. Methodology

The methodology selected to address this study is the perceived radiological risk profile. It is based on the use of variables and surveys. The survey questionnaire incorporated an informed consent form for all participants, ensuring they were fully apprised of the study’s objectives, methodology, potential risks, benefits, and their voluntary right to participate or withdraw at any juncture without adverse consequences. Moreover, the investigation was conducted with the formal approval of the Research Ethics Committee at the Instituto Nacional del Cáncer (INCART), adhering to established ethical guidelines for human subject research as outlined in international standards, such as the Declaration of Helsinki. One aspect linked to the use of this methodology is the availability of specialized software for these types of studies. The RISKPERCEP code, developed by an interdisciplinary group from the University of Havana, Cuba, was applied to multiple studies of occupational risk perception and public risk perception [42,43].

The questionnaire, developed especially for this study, was aimed at workers occupationally exposed to radiation from radiopharmaceutical production units. The survey was conducted between April and May 2024. The countries surveyed (thirteen) for this research were as follows: Argentina, Brazil, Chile, Colombia, Costa Rica, Cuba, Ecuador, Mexico, Panama, Peru, Portugal, Dominican Republic, and Venezuela. A total of 46 individuals directly involved in cyclotron-based radiopharmaceutical production were surveyed, including cyclotron operators, radiopharmacy managers, and radiation safety officers responsible for the facility. The algorithm used in the study is illustrated in Figure 1.

The design of risk perception variables depends on the objectives of the study [44,45,46]. For example, for the analysis of psychosocial risks, three types of variables are used (Table 1): those related to the individual, those related to the nature of the risk or physical risk, and those related to the management of the risk or risk managed [42,44]

Another important aspect regarding the selection of variables is the analysis of their relationship with the perception of risk associated with each one, detecting that some behave in a directly proportional way (direct behavior), such as the catastrophic potential, the panic generated, and the immediacy of the consequences, while others do so in an inverse way, such as familiarity with the risk, the ability to control it, and the reversibility of its consequences (inverse behavior). The variable risk compression has the particularity that its behavior, with respect to the associated risk perception, is extreme, which means that experts and non-specialists in the subject alike underestimate it (extreme behavior).

As a necessary simplification, to avoid adding subjectivity to the study, the variables considered are independent of each other, and each one has a similar contribution to the quantification.

Rules proposed by experts have been followed for the design of the survey [42,47,48]. The questionnaire must be adapted to the types of hazards and the study groups; it must generate empathy, and move from the known to the uncertain, from the general to the particular, and from the institutional to the individual [47]. To measure this perception, psychometric tools such as the Likert scale are used, which allows respondents to rate their level of agreement or evaluation against specific statements [49]. In this context, a 3-point Likert scale is used, where 1 represents underestimation of risk, 2 represents adequate perception, and 3 represents overestimation, and it was applied to a 30-question questionnaire answered by 46 workers.

As part of the refinement of the survey, it was submitted to a Delphi round of experts [50]. In addition, the survey passed a random response probability test using a Gaussian distribution. The data used were 30 questions, focusing on population, probability of success, and failure of each question, 0.4 and 0.6, respectively, and a value of randomly answered questions that were tested until an amount of 23 was reached. Given this amount, it was shown that a zero probability of random response was reached. Since a correct random answer of 23 questions is considered very unlikely, the quality of the survey is demonstrated by this indicator.

### 2.2. Statistical Analysis of the Questionnaire

A reliability analysis of the consistency of the risk perception questionnaire is carried out, which has 30 questions, evaluating individual, technological, and risk management variables. IBM SPSS Statistics version 31.0 is used for this purpose. As a result, a question (Do you work only in that center? linkage variable) was eliminated because it had zero variance. Table 2 shows the results of the reliability analysis using Cronbach’s alpha statistic.

A Cronbach’s alpha value of 0.718 is within the acceptable range (greater than 0.7), according to Kotian et al. [51], indicating that the questionnaire questions are reasonably consistent in measuring the predicted constructs. This suggests that the scale is reliable enough for exploratory research or preliminary conclusions. There is potential room for improvement to achieve good reliability (>0.8) through question optimization or a larger sample [52].

On the other hand, the standardized Cronbach’s alpha adjusts for differences in item variances by standardizing items (mean = 0, variance = 1) before calculating the reliability coefficient. This provides a more accurate estimate when questions have uneven variances. The increase to 0.760, an improvement of 0.042 over raw alpha, indicates that variability in responses to questions (e.g., due to different scales or response patterns) was reducing raw consistency. The standardized value is within the range of acceptable to good (>0.7), suggesting greater reliability when accounting for these differences [53].

Table 3 shows the results of the analysis of variance. The F statistic (11.342) measures the significant variance between the 29 questions (*p* < 0.001), supported by an acceptable value of η^2^ (0.183), indicating strong discriminant validity. The degrees of freedom (df) were 45 for “Between people”, 28 for “Between elements”, and 1260 for Residual. Acceptable ranges are met (*p* < 0.05 for F, df consistent with design).

Table 4 shows the analysis of variance for the individual variables (see Table 4).

The subscale of 11 questions corresponding to the individual variables is statistically acceptable for analysis. The high average of communality (0.658), six strong questions (> 0.7), and the contribution to variance of 59.871% explained by four components indicate that the subscale effectively captures aspects of individual knowledge and perception related to work with ionizing radiation.

Table 5 shows the analysis of variance for the variables associated with technological practice (see Table 5).

The statistical analysis indicates that the first six components, with a cumulative explained variance of 76.116%, are sufficient to construct the variable associated with the practice. This selection ensures a full representation of data variability, while adhering to standard PCA retention criteria. Finally, Table 6 shows the analysis for the variables associated with radiological risk management (see Table 6).

The statistical analysis indicates that the first three components, with a cumulative explained variance of 83.62%, are sufficient to construct the variables associated with risk management. This selection ensures a complete representation of data variability while adhering to standard PCA retention criteria. The inclusion of the fourth component is not necessary unless a full explanation of 100% of the variance is required, which is less common in practice due to diminishing returns.

## 3. Results and Discussion of the Risk Perception Study

Table 7 shows the demographic information of the participants of the risk perception survey. It can be observed that about 70% are men, and a similar percentage for the age group under 45 years old. More than half of those surveyed stated that they have postgraduate studies, which indicates that it is a job with high academic qualifications. From the perspective of wage income, it is a well-paid job.

The results for the mean value and its standard deviations are shown in Table A1 (Appendix B).

For a 30-question questionnaire answered by 46 workers, the total number of responses is 1380, and the overall average is calculated by adding up all the answers and dividing by the number of responses. Since each answer is on a scale of 1 to 3, the average will be between 1 and 3. To interpret this average and categorize the general behavior of the group, the range is divided into three approximately equal intervals, following common practices in the interpretation of Likert scales [54].

Likert scales are widely used measurement tools in various research domains, although their proper design and analysis remain a challenge. These scales typically use ordinal response options (e.g., 1–5 or 1–7 points) where responses can be ranked, but the distances between points are not necessarily equal [55]. Common interpretation practices involve dividing the scale range into roughly equal intervals to categorize responses [56,57]. Confusion around whether Likert data should be treated as ordinal or interval data continues to create analytical challenges [58]. These ranges are derived by dividing the total range of risk perception, which varies between 1 and 3 equal parts of about 0.67 each. An average between 1 and 1.67 indicates that the group tends to underestimate the risks, which could lead to unsafe practices. An average between 1.68 and 2.33 suggests an adequate perception, reflecting a balanced understanding of risks. An average between 2.33 and 3 indicates overestimation, which could lead to anxiety or rejection of necessary procedures [55]. On the other hand, [59] emphasizes that using responses to a single item as representative of a concept risks potentially misleading outcomes when selecting a single statement to represent a more complex outcome. Responses to individual items usually have a low degree of relationship with a composite score. This approach supports criteria-based interpretation, where thresholds are derived from the theoretical range of the scale, promoting consistency in fields such as psychology or education [60].

The graph of the perceived risk profile obtained from the subjective risk study among cyclotron workers in the Latin American area is shown in Figure 2.

This study also breaks down the behavior of risk perception by gender (Figure 3). The results of the survey on radiological risk perception in workers occupationally exposed to ionizing radiation in the production of cyclotron PET radiopharmaceuticals reveal notable differences between men and women in several psychometric dimensions, with implications for safety culture and compliance with protocols.

In general, women (n = 14) show higher averages in most variables (Figure 3), indicating a tendency to overestimate risk or maintain more adequate perceptions, which could reflect greater emotional fear or sensitivity to long-term impacts, which is consistent with the literature suggesting that women perceive greater radiological risks due to factors such as concern about hereditary effects (in contrast, men (n = 32) exhibit lower averages, with greater underestimation, encouraging complacency. These patterns, confirmed by the mean values and their variability, are then analyzed by variable, highlighting interventions based on IAEA regulations.

Several studies detail that female workers tend to report higher perceptions of radiological risk than their male counterparts. Whitney et al. [61] observed that pediatric anesthesiologists expressed 66% more concern about radiation exposure (*p* = 0.002). Liang et al. [62] found that 90% of female radiation therapists expressed concern about fetal effects compared to 47.83% of men (*p* < 0.001), while Drobov et al. [63] reported a significant association (β = 2.91, *p* < 0.001) between the female sex and elevated risk scores. This research describes the responses of women in terms of heightened emotional fear, anxiety about hereditary and long-term impacts, and a greater tendency to adjust work practices in accordance with IAEA safety protocols. On the contrary, several studies point to lower risk estimates among men, a pattern that may underlie a more accommodating adherence to protective measures. However, some studies, such as Younesi Heravi et al. [64] and Kyei et al. [65] found no significant gender differences, suggesting that context, background, or cultural factors may influence these effects.

In the same way, behavior by age group is included, as shown in Figure 4. Occupationally exposed workers in the field of PET radiopharmaceutical production report different perceptions of radiological risk according to age. In the present study, workers in the age range between 26 and 45 years (n = 31) obtained higher scores in the measures of risk severity, control capacity, and awareness than those aged 46 years or older (n = 15).

Several studies show that younger workers tend to overestimate risk or maintain more cautious perceptions, presumably because they are less habituated to chronic exposure, while older, more experienced workers underestimate risk, a pattern that can allow for complacency. For example, Hashiguchi et al. [66] and Moore et al. [67] explain that accumulated experience can attenuate risk sensitivity. On the contrary, Yashima and Chida [68] point out that younger technologists often show greater uncertainty regarding specific hazards. These age-related differences in psychometric dimensions have implications for safety culture and protocol compliance, with multiple reports advocating for targeted interventions aligned with the International Atomic Energy Agency guidelines.

### 3.1. General Analysis of Variables Outside the Range of Adequate Perception

The results of the survey on radiological risk perception in workers occupationally exposed to ionizing radiation in the production of cyclotron PET radiopharmaceuticals reveal patterns of underestimation and overestimation in key psychometric dimensions such as Familiarity (FAMI), Panic (PANI), Reversibility of Consequences (REVE), and Immediacy of Consequences (INME), with implications for the culture of safety and adherence to protocols. Two dimensions show means below two, indicating underestimation that fosters complacency and minimizes chronic exposures. Another exhibits an average slightly above two, suggesting overestimation that amplifies emotional “fear” and generates unnecessary anxiety. These findings, confirmed by confidence intervals and considering the standard error (SE) ranges of 0.0966 to 0.1053 for the underestimation dimensions and 0.1439 for the overestimation dimension as an argument for assessing the accuracy of the estimates, are discussed below in relation to the literature and contextual factors, highlighting IAEA policy-based interventions.

### 3.2. Familiarity (FAMI)

#### 3.2.1. Descriptive Interpretation of the Average and Variability

Average (1.7826): Clear tendency to underestimate, where perceived experience minimizes risks such as neutrons in a cyclotron or manipulation of PET isotopes.Standard deviation (SD = 0.8410): High variability (CV ≈ 47.2%), indicating diversity: novices might overestimate, and experts underestimate by habituation.Contextual implications: In PET, familiarity with routines (e.g., FDG dispensing) can generate complacency, increasing effective exposure.

#### 3.2.2. Inferential Analysis: Accuracy and Generalization

Standard error (SE = 0.124): Moderate accuracy.95% confidence interval: (1.533, 2.032), focused on underestimation.Limitations: High SD suggests influenced subgroups.

#### 3.2.3. Comparisons with the Literature and Influencing Factors

Studies on radiation risk perception reveal significant gaps between actual and perceived risks among workers and the public. Studies show that familiarity with radiation work can lead to an underestimation of risks, with experienced workers showing a lower perception of risk [27,69]. Rincón et al. [70] found that workers with more than 20 years of radiation experience recognized safety principles but struggled with the risk–benefit justification. The relationship between knowledge and risk perception is complex: while increased knowledge of radiation correlates with a lower perception of the risk of low-dose exposure [27], this can potentially lead to complacency. [71] emphasizes the distinction between risk estimation and risk perception, noting that scientific data often do not align with patient concerns. [72] highlights how radiation risks are frequently misunderstood, and perceived risks are inconsistent with actual statistical risks. Rehani [73] proposes that severity and latency factors significantly influence risk perception, suggesting the need for more rational approaches to risk communication.

### 3.3. Panic (PANI)

#### 3.3.1. Descriptive Interpretation of the Mean and Variability

Average (1.8261): Indicates an almost adequate perception, but with a slight tendency to underestimate risk. Workers perceive the feelings of panic associated with ionizing exposure (e.g., fear of high doses during cyclotron failure) as less intense than real, possibly due to habituation or reliance on protections, minimizing emotional dread.Standard deviation (SD = 0.9956): High variability (CV ≈ 54.5%), suggesting heterogeneity: some strongly underestimate (about 1, low perceived panic), others see greater anxiety (about 2–3), possibly due to differences in personal experience or knowledge of risks.Contextual implications: In PET cyclotrons, underestimation of panic can lead to complacency in stressful situations, such as emergencies, affecting emotional response and adherence to safety protocols, although it reduces unnecessary chronic stress.

#### 3.3.2. Inferential Analysis: Precision and Generalization

Standard error (SE = 0.1468): Moderate accuracy, reflecting uncertainty due to high SD.95% confidence interval: (1.530, 2.122), crossing 2 but centered down, confirming slight underestimation in similar groups.Limitations: n = 46; high SD indicates possible asymmetry or subgroups influenced by psychological factors.

#### 3.3.3. Comparisons with the Literature and Influencing Factors

Research on occupational radiological risk perception reveals complex psychological dynamics among exposed workers. Studies demonstrate that “dread” and severity feelings significantly influence risk perception, with workers often underestimating expert knowledge while overvaluing their personal risk assessment [74]. Context variables such as dread and personal control are important predictors of perceived risk seriousness [75]. Workers with greater experience in ionizing radiation exposure show lower risk perception, suggesting habituation effects reduce anxiety over time [76]. The IAEA emphasizes managing psychological aspects like anxiety to calibrate exposure perceptions [70].

### 3.4. Reversibility of Consequences (REVE)

#### 3.4.1. Descriptive Interpretation of the Average and Variability

Average (1.6087): It indicates a clear trend towards underestimation of risk. Workers perceive the consequences of ionizing exposure (e.g., deterministic effects such as burns or stochastic effects such as cancer) as more reversible than real, possibly due to reliance on medical treatments or minimization of irreversible long-term damage.Standard deviation (SD = 0.7142): Moderate–high variability (CV ≈ 44.4%), suggesting heterogeneity: some strongly underestimate (about 1, high perceived reversibility), others see lower reversibility (around 2–3), possibly due to differences in knowledge of biological effects or personal experience.Contextual implications: In PET cyclotrons, underestimation of irreversibility can lead to complacency in the face of chronic exposures, ignoring permanent damage such as genetic mutations or cancer, affecting adherence to dose limits.

#### 3.4.2. Inferential Analysis: Accuracy and Generalization

Standard error (SE = 0.1053): Moderate precision, indicating reasonable estimation despite variability.95% confidence interval: (1.396, 1.821), centered below 2, confirming dominant underestimation in similar populations.Limitations: n= 46; moderate SD suggests individual influences not captured.

#### 3.4.3. Comparisons with the Literature and Influencing Factors

Research on occupational radiological risk perception reveals significant gaps between the experts’ and the public’s understanding of radiation hazards. Studies show that occupationally exposed workers often have different perceptions of risk compared to the general population, with experts showing a higher perception of medical X-rays and natural radiation, but a lower perception of nuclear waste and accidents [18]. Workers with extensive experience recognize the ALARA principles based on distance, time, and shielding in accordance with IAEA guidelines, although justification remains problematic, with 30% of procedures being unjustified [70]. The relationship between absorbed dose and risk is poorly understood, particularly for low doses, despite workers’ knowledge of the nature of radiation and health effects [77]. Epidemiological studies of occupationally exposed groups provide valuable evidence for radiation protection, particularly with respect to prolonged low-level exposure [78]. The IAEA emphasizes capacity building in radiobiological sciences and biological dosimetry to strengthen understanding of the effects of radiation [79].

### 3.5. Immediacy of Consequences (INME)

#### 3.5.1. Descriptive Interpretation of the Average and Variability

Average (2.2609): Indicates an almost adequate perception, but with a slight tendency to overestimate risk. Workers perceive the consequences of ionizing exposure (e.g., immediate deterministic effects such as erythema or long-term stochastic effects) as more immediate than real, possibly because of the “fear” associated with radiation, amplifying the perceived urgency.Standard deviation (SD = 0.9760): High variability (CV ≈ 43.2%), suggesting heterogeneity: some strongly overestimate (about 3, very immediate consequences), others perceive higher latency (towards 1–2), possibly due to differences in knowledge of biological effects.Contextual implications: In PET cyclotrons, overestimation of immediacy may generate anxiety or excessive caution, although useful for acute risks; however, it could underestimate chronic effects such as cancer, affecting the management of cumulative exposures.

#### 3.5.2. Inferential Analysis: Accuracy and Generalization

Standard error (SE = 0.1439): Moderate accuracy, reflecting uncertainty due to high SD.95% confidence interval: (1.971, 2.551), crossing 2 but centered above, confirming slight overestimation in similar groups.Limitations: n = 46; high SD indicates possible asymmetry or influenced subgroups.

#### 3.5.3. Comparisons with the Literature and Influencing Factors

People’s perception of radiation risks, including genetic effects and cancer, often exceeds actual exposure levels, leading to increased anxiety and depressive symptoms. Distrust of authorities and inadequate communication further intensify these perceptions [80]. IAEA highlights communication of times of consequences to calibrate perceptions of occupational exposure [81].

Factors: Knowledge of biological effects (high perceived immediacy overestimates) is argued by articles demonstrating how limited understanding amplifies immediate fear in radiological risks [82]. Research has shown that a limited understanding of radiation biology significantly amplifies fear and overestimation of radiological risks. Brooks et al. [83] argue that fear of radiation has intensified since the atomic age and often lacks a scientific basis, with a clear understanding of the effects of radiation being crucial to alleviating such fears. This relationship between knowledge and fear is supported by Beauchamp [84] who demonstrated that fear levels are inversely correlated with understanding radiation among healthcare providers, noting that a lack of understanding leads to unreasonable radiophobia in both the public and medical staff. Feinendegen and Cuttler [85] identify that an inadequate overview of relevant data contributes to the “confused fear” of ionizing radiation among the public. The impact of training variability is evident in healthcare settings, where [86] emphasize that most medical personnel lack familiarity with radiation victims, but adequate knowledge and exercise can reduce radiation-related terror and improve community response capabilities.

### 3.6. Radiopharmaceutical Production in Latin America: Actions and Policies to Address Risk Perception

Psychometric analysis of risk perception among workers in the production of PET radiopharmaceuticals in Latin America reveals deviations from appropriate levels, with underestimation encouraging complacency in areas such as awareness of chronic effects and institutional reliability, and overestimation amplifying emotional responses to immediate threats. These gaps, exacerbated by demographic factors such as age and gender, highlight gaps in training, communication, and follow-up that undermine job security. To modulate perceptions towards a balanced consciousness, the following 15 global and regional actions and policies are proposed, based on established frameworks of key organizations. These include the International Atomic Energy Agency (IAEA), the International Labour Organization (ILO), the Ibero-American Forum of Radiological and Nuclear Regulatory Bodies (Grupo FORO), the Latin American Association of Medical Physics (ALFIM), and the Federation of Radiation Protection of Latin America and the Caribbean (FRALC). Implementation at the regional level can be supported by the FORO Group and FRALC for harmonization across Latin America, ensuring culturally adapted delivery through collaborative networks:Mandatory annual refresher training on radiation biology: Establish comprehensive annual courses that focus on the long-term biological impacts of ionizing radiation, including practical modules to improve understanding of the outcomes of chronic exposure. This action addresses gaps in the recognition of persistent effects by providing evidence-based education, with guidance from the IAEA on curriculum design and ALFIM support for regional adaptation in Latin American medical physics programs.Real-time dosimetry and feedback systems: Implementation of automated dosimetry tools with immediate warning mechanisms in cyclotron facilities to provide continuous exposure data, helping workers recognize common risks without relying on retrospective assessments. ILO standards for occupational monitoring can ensure implementation, while FRALC can facilitate the regional exchange of best practices to overcome awareness gaps in routine operations.Gender-sensitive risk communication workshops: Organizing workshops tailored to demographic differences, incorporating interactive sessions on emotional responses to radiation hazards to balance heightened concerns without inducing undue stress. The IAEA’s safety culture principles can guide the content, and the FORO Group coordinates delivery in all Latin American countries to address variations in perceptual biases.Age-stratified mentoring programs: pairing young and older workers in mentoring schemes to exchange experiences on radiation safety, foster a shared understanding of exposure dynamics, and reduce generational perceptual disparities. ALFIM’s educational frameworks can support program design, while ILO conventions on workplace equity ensure inclusive participation.Institutional transparency and auditing mechanisms: Conducting regular independent audits of security protocols with public reports to build credibility in regulatory bodies, filling gaps in perceived reliability through verifiable compliance data. The IAEA’s General Safety Requirements can provide the audit template, and FRALC advocates for regional standardization in Latin America.Virtual Reality Simulations for Temporary Risk Training: Utilization of virtual reality tools to simulate exposure scenarios, illustrating the timeline of radiation effects to correct misperceptions of urgency and promote accurate awareness of risk time. The methodologies of the AAPM working groups can inform the development of simulations, with the support of the FORO Group for their dissemination in Ibero-American networks.Task rotation policies for exposure management: Introduction of mandatory rotation programs for high-risk tasks to avoid normalization of mandatory exposures, ensuring that workers experience varied demands and maintain greater vigilance. ILO guidelines on occupational health can enforce these policies, and ALFIM helps to adapt them to the roles of medical physics in Latin America.Psychological resilience training programs: Offering specialized training in emotional management techniques for radiation-related anxiety, including coping strategies to balance responses to perceived threats. IRPA’s principles on stakeholder engagement can underpin the program, while FRALC can facilitate regional workshops tailored to Latin American contexts.Sharing regional databases of historical incidents: Creating a shared database of past radiation incidents with anonymous case studies for ongoing review, enhancing the collective memory of hazards without inducing fear. The IAEA’s SAFRON system can serve as a model, with the support of the FORO Group for Ibero-American integration.Multidisciplinary team-building exercises: Conducting team exercises involving physicists, technicians, and supervisors to strengthen safe collaborative practices, addressing gaps in risk awareness at the group level. AAPM’s quality management reports can guide exercise design, and ALFIM promotes their use in Latin American medical facilities.Evidence-based campaigns on hereditary effects: Launch of information campaigns using verified data on long-term hereditary impacts, distributed through digital platforms to reach diverse groups of workers. IAEA safety reports can provide the evidence base, while FRALC can coordinate regional dissemination.Supervisor leadership development courses: Supervisor training in proactive safety communication and supervision, ensuring consistent manifestation of protective measures across all teams. IRPA’s guiding principles can inform course content, with Grupo FORO facilitating cross-border training in Latin America.Workload assessment policies: Implementation of periodic assessments of work demands with adjustments for exposure intensity, using standardized tools to highlight unavoidable risks. ILO conventions on workplace safety may require them, with ALFIM supporting specific assessments of physics.Integrated safety culture audits: Conducting holistic audits combining perception surveys with performance metrics to identify and correct imbalances, ensuring continuous modulation of awareness. The IAEA’s safety culture assessment guidelines can provide the framework, and FRALC advocates for regional benchmarking in Latin America.

These actions and policies aim to close perceptual gaps by promoting evidence-based awareness and a culture of safety, ultimately aligning with the Sustainable Development Goals in Latin America’s radiation practices.

## 4. Conclusions

This study provides a pioneering regional assessment of risk perception in medical cyclotron operations to produce PET radiopharmaceuticals in 13 Latin American countries, revealing nuanced deviations that report specific safety improvements. The use of a validated questionnaire (Cronbach’s α > 0.7), statistical analysis of means, standard deviations (SD), standard errors (EE), 95% confidence intervals (t-Student distribution) and coefficients of variation (CV) identified underestimation in dimensions such as consequence reversibility (mean 1.61, SD 0.7142, SE 0.1053, CV ≈ 44.4%) and familiarity (mean 1.78, SD 0.8410, SE 0.124, CV ≈ 47.2%), encouraging complacency towards chronic exposures, together with overestimation of the immediacy of the consequences (mean 2.26, SD 0.9760, SE 0.1439, CV ≈ 43.2%), amplifying emotional responses. Stratifications by gender and age further clarify patterns: women and younger workers (26–45 years) show a greater overestimation in immediacy and generational effects, while men and older workers (≥46 years) show a greater underestimation in reversibility and institutional trust, underscoring demographic influences on perceptual biases.

Overall, these findings indicate a predominantly underestimated risk profile that may compromise adherence to dose optimization, in contrast to selective overestimation that drives unnecessary surveillance. Policies to modulate perception include mandatory annual refresher courses in biology, real-time dosimetric feedback, and gender- and age-appropriate communication campaigns, in line with the IAEA GSR Part 3 and SSG-46 for sustainable protection. Opportunities for future studies encompass longitudinal evaluations with larger cohorts to track post-intervention perception changes and comparative analyses across regions of the world to refine Latin American strategies. Comparative studies in developed regions are needed to assess whether economic and infrastructural differences influence risk perception; our review found no equivalent research outside Latin America.

## Figures and Tables

**Figure 1 ijerph-22-01885-f001:**
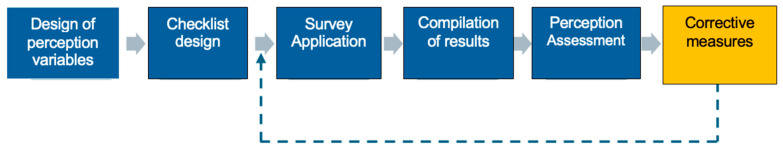
Algorithm for the study of risk perception.

**Figure 2 ijerph-22-01885-f002:**
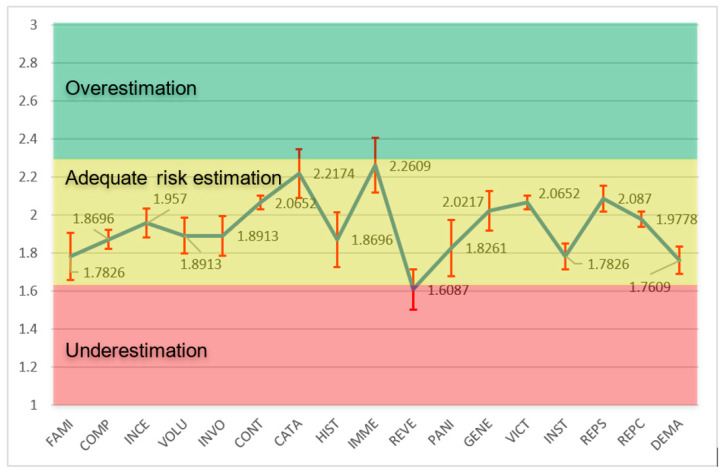
Perceived risk profile in radiopharmaceutical production (FAMI—Worker’s level of experience in the production of radiopharmaceuticals, COMP—Understanding of risk, INCE—Uncertainty, VOLU—Voluntariness, INVO—Personal involvement, CONT—Ability to control, CATA—Catastrophic potential, HIST—Past history of disasters or hazards, IMME—Immediacy of disasters/consequences, REVE—Reversibility of consequences, PANI—Panic, GENE—Effect on generations, VICT—Identity of victims, INST—Trust in institutions, REPS—Supervisors’ response, REPC—Peer response, and DEMA—Labor lawsuit). The dimensionless risk perception scale (Y-axis) indicates the following: (1–1.67)—underestimation of risk; (1.68–2.23)—adequate risk estimation; and (2.24–3)—Overestimation of risk.

**Figure 3 ijerph-22-01885-f003:**
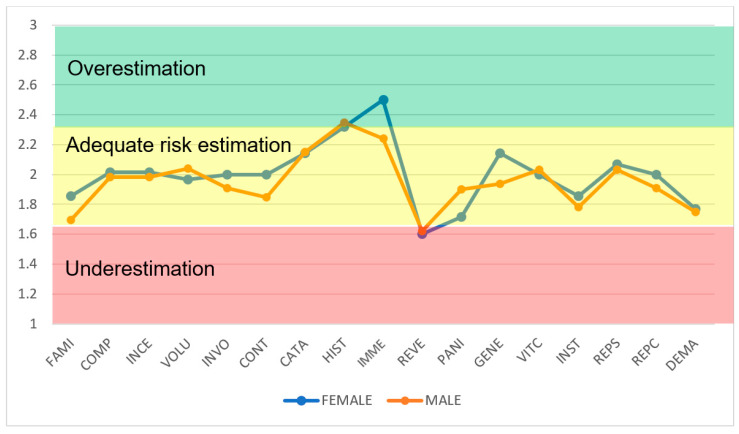
Perceived risk profile in cyclotron areas by gender (FAMI—Worker’s level of experience in the production of radiopharmaceuticals, COMP—Understanding of risk, INCE—Uncertainty, VOLU—Voluntariness, INVO—Personal involvement, CONT—Ability to control, CATA—Catastrophic potential, HIST—Past history of disasters or hazards, IMME—Immediacy of consequences, REVE—Reversibility of consequences, PANI—Panic, GENE—Effect on generations, VICT—Identity of victims, INST—Trust in institutions, REPS—Supervisors’ response, REPC—Colleagues’ response, and DEMA—Labor lawsuit). The dimensionless risk perception scale (Y-axis) indicates the following: (1–1.67)—underestimation of risk; (1.68–2.23)—adequate risk estimation; and (2.24–3)—Overestimation of risk.

**Figure 4 ijerph-22-01885-f004:**
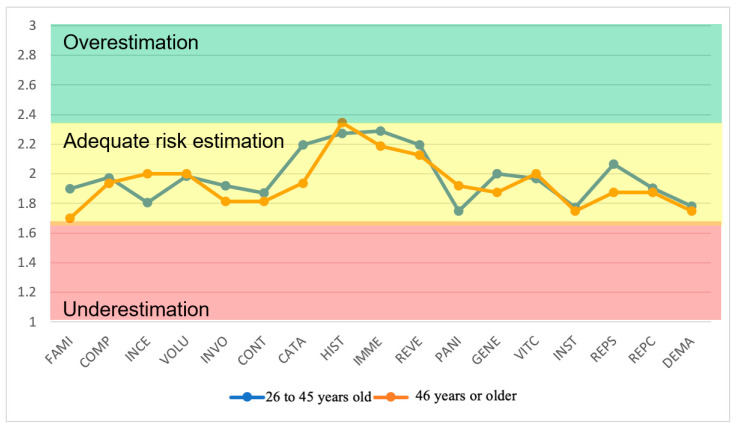
Perceived risk profile in cyclotron areas by age groups (FAMI—Worker’s level of experience in the production of radiopharmaceuticals, COMP—Understanding of risk, INCE—Uncertainty, VOLU—Voluntariness, INVO—Personal involvement, CONT—Ability to control, CATA—Catastrophic potential, HIST—Past history of disasters or hazards, IMME—Immediacy of consequences, REVE—Reversibility of consequences, PANI—Panic, GENE—Effect on generations, VICT—Identity of victims, INST—Trust in institutions, REPS—Supervisors’ response, REPC—Colleagues’ response, and DEMA—Labor lawsuit). The dimensionless risk perception scale (Y-axis) indicates the following: (1–1.67)—underestimation of risk; (1.68–2.23)—adequate risk estimation; and (2.24–3)—Overestimation of risk.

**Table 1 ijerph-22-01885-t001:** Variables used for the study of risk perception in occupationally exposed workers in radiopharmacy units.

No.	Variable (Code) ^1^ [Quiz Questions ^2^]	Description
**Individual Variables**		
1	Familiarity (FAMI) [1]	Degree of experience of the worker in the production of radiopharmaceuticals.
2	Understanding Risk (COMP) [2,3,4,5,6]	Degree of knowledge of the individual about the radiological risk.
3	Uncertainty (INCE) [7]	The subject’s perception of the degree of knowledge that science has about radiological risk.
4	Voluntarism (VOLU) [8,9]	Degree of decision by the subject as to whether to expose himself to radiological risk.
5	Personal involvement (INVO) [10,11]	The degree to which occupational exposure directly affects him or his family (target of risk).
6	Ability to control (CONT) [12]	Degree to which the subject can perform an effective conduct to modify the situation of radiological risk.
7	Employment (VINC) [13]	The degree to which the individual depends for his or her subsistence on the performance of the work related to the radiological risk.
**Variables of a physical nature**		
8	Catastrophic potential (CATA) [14,15]	Degree of the fatality of the consequences of radiological exposure and its concurrence in space and time.
9	History of accidents (HIST) [16,17]	Degree to which the production of radiopharmaceuticals has a prior history of catastrophes or hazards
10	Immediacy of consequences (INME) [18]	The degree to which the consequences of occupational radiological exposure are immediate.
11	Reversibility of consequences (REVE) [19,20]	Extent to which the consequences of occupational radiological exposure are reversible
12	Panic (PANI) [21]	The degree to which radiological risk produces sensations such as fear, terror or anxiety.
13	Effect on generations (GENE) [22,23,24]	The degree to which occupational radiological exposure can affect future generations.
14	Identity of the victims (VICT) [25]	The degree to which occupational exposure has affected people close to them or is only measured in the form of statistics.
**Variables related to risk management**		
15	Trust in Institutions (INST) [26,27]	The degree to which the worker trusts or gives credibility to the institutions responsible for radiation safety.
16	Supervisors’ response (REPS) [28]	Degree to which supervisors express themselves regarding the radiological protection of personnel.
17	Response from colleagues (REPC) [29]	Degree to which the conduct of the work group supports safe attitudes of practice with ionizing radiation
18	Labor Demand (DEMA) [30]	Degree to which working conditions demand exposure to radiological hazards.

^1^ the variables are related to a code for their computer identification; ^2^ The questionnaire is presented in Appendix A.

**Table 2 ijerph-22-01885-t002:** Reliability analysis of the questionnaire by Cronbach’s alpha.

Reliability Statistics		
Cronbach’s alpha	Cronbach’s alpha based on standardized items	Number of questions passed
0.718	0.760	29

**Table 3 ijerph-22-01885-t003:** Analysis of variance of the questionnaire.

ANOVA						
		Sum of Squares	df	Middle Square	F	Sig
Between people		41.301	45	0.918		
Inside people	Between elements	82.186	28	2.935	11.342	<0.001
	Residual	326.090	1260	0.259		
	Total	408.276	1288	0.317		
Total		449.577	1333	0.337		

**Table 4 ijerph-22-01885-t004:** Analysis of variance of individual variables.

ANOVA						
Component	Initial Eigenvalues			Load Extraction Sums Squared		
	Total	% Variance	Cumulative %	Total	% Variance	Cumulative %
1	2.416	21.964	21.964	2.416	21.964	21.964
2	1.730	15.724	37.688	1.730	15.724	37.688
3	1.317	11.971	49.659	1.317	11.971	49.659
4	1.123	10.211	59.871	1.123	10.211	59.871
5	0.994	9.032	68.903			
6	0.920	8.364	77.267			
7	0.751	6.827	84.094			
8	0.697	6.332	90.427			
9	0.443	4.031	94.458			
10	0.311	2.827	97.284			
11	0.299	2.716	100.000			
Extraction Method: Principal Component Analysis (PCA)						

**Table 5 ijerph-22-01885-t005:** Analysis of variance by groups of variables for the variables associated with technological practice.

ANOVA						
Component	Initial Eigenvalues			Load Extraction Sums Squared		
	Total	% Variance	Cumulative %	Total	% Variance	Cumulative %
1	2.287	19.059	19.059	2.287	19.059	19.059
2	2.031	16.922	35.981	2.031	16.922	35.981
3	1.431	11.921	47.902	1.431	11.921	47.902
4	1.220	10.165	58.067	1.220	10.165	58.067
5	1.144	9.533	67.600	1.144	9.533	67.600
6	1.022	8.516	76.116	1.022	8.516	76.116
7	0.854	7.117	83.233			
8	0.744	6.197	89.431			
9	0.501	4.174	93.605			
10	0.347	2.895	96.500			
11	0.281	2.345	98.845			
12	0.139	1.155	100.000			
Extraction Method: Principal Component Analysis (PCA).						

**Table 6 ijerph-22-01885-t006:** Analysis of variance by groups of variables for variables associated with risk management.

ANOVA						
Component	Initial Eigenvalues			Load Extraction Sums Squared		
	Total	% Variance	Cumulative %	Total	% Variance	Cumulative %
1	1.415	35.385	35.385	1.415	35.385	35.385
2	1.101	27.536	62.921	1.101	27.536	62.921
3	0.828	20.698	83.619			
4	0.655	16.381	100.000			
Extraction method: principal component analysis (PCA)						

**Table 7 ijerph-22-01885-t007:** Demographic information of the respondents.

Variable	Classification	Number	Proportion
Gender	Female	14	30.4%
	Male	32	69.6%
Age	26–45 years	31	67.4%
	>45 years old	15	32.6%
Level of Education	Technician	5	10.9%
	University	17	37.0%
	With postgraduate degrees	24	52.1%
Income from salary	Lower than the national average	14	30.5%
	Adequate	21	45.6%
	Higher than the national average	11	23.9%

## Data Availability

The data presented in this study are available on request from the corresponding author.

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
