# Peer review of "Pilot Study on Risk Perception in Practices with Medical Cyclotrons in Radiopharmaceutical Centers in Latin American Countries: Diagnosis and Corrective Measures"

_ijerph, 2025, doi:10.3390/ijerph22121885_

Round 1
Reviewer 1 Report
Comments and Suggestions for Authors
The article examines a study in which questions were posed to workers at PET radiopharmaceutical production centers in 14 Latin American countries and Portugal. The answers were ranked and the workers' perceptions of various factors were examined. The article is well-written in terms of method and conclusions. However, I have many doubts about the sample used, as I believe it examines very different situations. First of all, it's unclear how many radiopharmaceutical production centers there are in these countries. One per nation? It seems strange to me that a country as large as Brazil has only one. The questionnaire was administered to 46 workers. This seems like a very low number. Furthermore, what role do these workers play in these centers? It's important to have this data as well. The countries surveyed have very different governments and laws. Is the radiation protection situation homogeneous? It seems strange to me that this is the case. I believe the questionnaire should be extended to a larger number of workers and the above elements examined.
Comments on the Quality of English LanguageThe English language is fluent and easy to read.
Author Response
- Addition of quantitative data on PET production centers
A new reference has been incorporated into the Introduction section (page 2, after line 58) to provide regional context on the infrastructure for PET radiopharmaceutical production. The following sentence was added:
In the Latin American and Caribbean region, a total of 67 cyclotron facilities dedicated to PET radioisotope production have been identified, of which 54 are currently operational [6], highlighting the significant expansion of this technology despite ongoing challenges in radiation protection and regulatory harmonization.
The new reference is:
[6] Ávila-Rodríguez MA, Manríquez R, Contreras A, Solís C, Zarza V, Morales Ávila E, et al. Cyclotrons and radiopharmaceutical production in Latin America and the Caribbean: A survey on infrastructure and regulatory framework. Appl Radiat Isot. 2022;180:110048. https://doi.org/10.1016/j.apradiso.2021.110048
This addition strengthens the regional framing of the study and supports the relevance of the sample (46 workers from 13 countries) relative to the existing infrastructure.
2. Clarification of the survey target population
The Materials and Methods section (subsection 2.1, page 5) has been revised to explicitly define the surveyed population. The original sentence:
A total of 46 individuals related to the production of radiopharmaceuticals with cyclotrons were interviewed.
has been replaced with:
A total of 46 individuals directly involved in cyclotron-based radiopharmaceutical production were surveyed, including cyclotron operators, radiopharmacy managers, and radiation safety officers responsible for the facility.
This clarification ensures transparency regarding the professional roles of respondents, all of whom are occupationally exposed and directly responsible for operational and safety practices.
- Regulatory harmonization and IAEA standards
To address potential concerns about variability in national regulations, the following sentence has been added to the Introduction (page 2, after the new infrastructure sentence):
Although national regulatory frameworks vary, the International Atomic Energy Agency (IAEA) standards—particularly GSR Part 3 [7] and SSG-46 [8]—promote harmonization of good manufacturing practices (GMP) and radiation safety in radiopharmaceutical production across member states, providing a common technical and safety baseline for the region.
Corresponding references added:
[7] International Atomic Energy Agency. Radiation Protection and Safety of Radiation Sources: International Basic Safety Standards (GSR Part 3). Vienna: IAEA; 2014.
[8] International Atomic Energy Agency. Radiation Safety in Industrial Radiography and in the Production of Radiopharmaceuticals (SSG-46). Vienna: IAEA; 2018.
This addition acknowledges regulatory diversity while emphasizing that IAEA guidelines serve as a unifying framework, reducing heterogeneity in safety culture and operational standards among the surveyed centers.
These changes enhance the contextual depth, methodological clarity, and regulatory grounding of the study, directly responding to the reviewer’s concerns.
Reviewer 2 Report
Comments and Suggestions for Authors
The structure of the manuscript, with some technical reservations, is correct because it contains all the necessary elements. The authors also explain in detail the importance and role of each statistical tool used to analyze the data, and selected material was subjected to tests to verify the reliability of the surveys. Overall, the survey results indicate that employees underestimate risk. Therefore, identifying ways to improve this situation is a valuable part of this work..
I have the following comments regarding some of the statements.
- Citing the literature, the authors indicate that experience and knowledge support caution regarding the risks associated with ionizing radiation, while on the other hand, those educated persons in this field reduce the importance of threats posed by, for example, nuclear waste and radiation accidents. Can this be commented on ?
- Lines 163-169.
The authors point out that the surveys collected opinions from people from 14 countries, but only 13 were mentioned.
Forty-six people participated in the study and their number is rather low. It seems useful to provide the total number of people in these countries involved in the production of radiopharmaceuticals using cyclotrons.
- Lines 343-345.
At the beginning (lines 108-109), the authors noted that older workers, due to their extensive decision-making experience, tend to use highly developed strategies to make effective decisions in real-world situations. The current research shows that younger workers demonstrated greater risk awareness than older ones. As the authors later point out, these results vary widely among different authors. Could these differences be due to uncertainty related to the size of the datasets?
- The economic situation of society may influence the results, as the authors of the manuscript point out. Would it be possible to compare the current results with similar studies conducted in a one of the highly developed country ?
Author Response
- Experience/knowledge and risk perception
The apparent contradiction reflects context-specific effects: greater knowledge reduces perceived risk for low-dose/chronic exposures (e.g., nuclear waste [23,24]), but increases caution in acute, operational settings like cyclotron facilities. In our study, familiarity with short-lived isotopes (e.g., ¹⁸F, t½ = 109.8 min) promotes underestimation due to routine handling and rapid decay, reducing perceived immediacy and dread.
- Number of countries and sample size (lines 163–169)
Corrected: 13 countries were surveyed
The survey was conducted between April and May 2024 in 13 countries (Argentina, Brazil, Chile, Colombia, Costa Rica, Cuba, Ecuador, Mexico, Panama, Peru, Portugal, Dominican Republic and Venezuela.).
Sample size (n=46) is appropriate given 67 regional cyclotrons (54 operational) [6] and targeted roles (cyclotron operators, radiopharmacy managers, radiation safety officers). Total personnel estimate: ~3–5 per facility → ~200–300 regionally. Our sample covers ~15–23% of key personnel across 13 nations.
- Age-related differences (lines 343–345 vs. lines 108–109)
Differences may reflect small subgroup sizes (26–45 years: n=28; ≥46 years: n=18) and contextual factors (e.g., younger workers more exposed to recent training). Variability may stem from limited subgroup sizes and differing exposure to updated safety protocols, with younger workers potentially receiving more recent IAEA-aligned training.
- Comparison with developed countries
No similar studies exist for cyclotron workers in high-income settings. Added to Conclusions (new final sentence):
Comparative studies in developed regions are needed to assess whether economic and infrastructural differences influence risk perception; our review found no equivalent research outside Latin America.
Round 2
Reviewer 1 Report
Comments and Suggestions for Authors
The authors answered the questions posed. However, in my opinion, the number of workers surveyed remains low. I would suggest noting somewhere in the paper, maybe in the title, that this is a pilot study and submitting a follow-up study within two years, with a sample size at least four times larger.
Author Response
Reviewer comment: The authors answered the questions posed. However, in my opinion, the number of workers surveyed remains low. I would suggest noting somewhere in the paper, maybe in the title, that this is a pilot study and submitting a follow-up study within two years, with a sample size at least four times larger.
Author response: We agree with the reviewer's assessment of the sample size (n=46) as a limitation and recognize that framing the work as a pilot study enhances transparency, particularly given its exploratory and pioneering nature as the first regional assessment in Latin America.
Furthermore, we plan to conduct a new study in the future to evaluate the impact of the proposed corrective measures on risk perception and safety practices, incorporating a larger number of PET radiopharmaceutical production centers across the region.
These changes are highlighted in yellow in the revised manuscript.